# Node Perturbation Can Effectively Train Multi-Layer Neural Networks

## Abstract

Backpropagation (BP) remains the dominant and most successful method for training parameters of deep neural network models. However, BP relies on two computationally distinct phases, does not provide a satisfactory explanation of biological learning, and can be challenging to apply for training of networks with discontinuities or noisy node dynamics. By comparison, node perturbation (NP), also known as activity-perturbed forward gradients, proposes learning by the injection of noise into network activations, and subsequent measurement of the induced loss change. NP relies on two forward (inference) passes, does not make use of network derivatives, and has been proposed as a model for learning in biological systems. However, standard NP is highly data inefficient and can be unstable due to its unguided noise-based search process. In this work, we develop a modern perspective on NP by relating it to the directional derivative and incorporating input decorrelation. We find that a closer alignment with directional derivatives together with input decorrelation at every layer theoretically and practically enhances performance of NP learning with large improvements in parameter convergence and much higher performance on the test data, approaching that of BP. Furthermore, our novel formulation allows for application to noisy systems in which the noise process itself is inaccessible, which is of particular interest for on-chip learning in neuromorphic systems.

## 1 Introduction

Backpropagation (BP) is the workhorse of modern artificial intelligence. It provides an efficient way of performing multi-layer credit assignment, given a differentiable neural network architecture and loss function (Linnainmaa, 1970). Despite BP's successes, it requires an auto-differentiation framework for the backward assignment of credit, introducing a distinction between a forward, or inference phase, and a backward, or learning phase, increasing algorithmic complexity and impeding implementation in (neuromorphic) hardware (Kaspar et al., 2021; Zenke and Neftci, 2021). Furthermore, BP has long been criticized for its lack of biological detail and plausibility (Grossberg, 1987; Crick, 1989; Lillicrap et al., 2016), with significant concerns again being the two separate forward and backward phases, but also its reliance on gradient propagation and the non-locality of the information required for credit assignment.

Alternative algorithms have been put forth over the years, though their inability to scale and difficulty in achieving levels of performance comparable to BP have held back their use. Many focus either on greedy local learning rules or on random feedback matrices and projections, which are deemed more plausible than symmetric feedback weights (Dellaferrera and Kreiman, 2022; Frenkel et al., 2021; Kohan et al., 2018; Hinton, 2022). The category of algorithms we focus on in this work are perturbation algorithms, specifically variants of node perturbation (NP) (Dembo and Kailath, 1990; Cauwenberghs, 1992), which require no feedback connectivity at all. In NP, node activations are perturbed by a small amount of random noise. Weights are then updated to produce the perturbed activations in proportion to the degree by which the perturbation improved performance. This method requires a measure of the loss function being optimized on a network both before and after the inclusion of noise. Such an approach is appealing because it leverages the same forward pass twice, rather than relying on two computationally distinct phases. It also does not require non-local information other than the global performance signal.

Despite these benefits, Hiratani et al. (2022) demonstrate that NP is extremely inefficient compared to BP, requiring two to three orders of magnitude more training cycles, depending on network depth and width. In addition, they found training with NP to be unstable, in many cases due to exploding weights. Another phenomenon uncovered in their work is that the covariance of NP's updates is significantly higher than that of BP, which can mathematically be described as an effect mediated by correlations in the input data.

In this work, we put forward three contributions. First, we reframe the process of node perturbation together with the subsequent measurement of the output loss change in terms of directional derivatives. Directional derivatives have been related to perturbation-based learning before in the context of forward gradient learning (Baydin et al., 2022; Ren et al., 2022) but never used to fully explicate a learning rule in this manner. This provides a more solid theoretical foundation for node perturbation on the whole. Second, we introduce an approximation, referred to as activity-based node perturbation, which is more efficient and has the additional advantage that it can be implemented in noisy systems such as imprecise hardware implementations (Gokmen, 2021) or biological systems (Faisal et al., 2008), where the noise itself is not directly measurable. Third, we propose to use a decorrelation method, first described by Ahmad et al. (2023), to de-bias activations. Because NP-style methods directly correlate perturbations of unit activities with changes in a reward signal, decorrelation of these unit activities helps to eliminate confounding effects. That is, it makes credit assignment more straightforward, effectively aligning gradient updates more closely with the natural gradient (Ahmad, 2024).

By combining decorrelation with the different NP methods, we find that it is possible to achieve orders of magnitude increase in model convergence speed, with performance levels rivaling networks trained by BP in certain contexts. Moreover, NP is a suitable candidate for learning on (noisy, resource-constrained) devices as it removes the need to implement the backward pass required for backpropagation. This is of particular interest for neuromorphic computing applications that make use of mixed-signal unconventional computing paradigms.

## 2 Methods

### 2.1 Node perturbation and its formulations

Let us define the forward pass of a fully-connected neural network, with $L$ layers, such that the output of a given layer, $l \in 1, 2, \ldots, L$ is given by $\mathbf{x}_l = f(\mathbf{a}_l)$, where $\mathbf{a}_l = \mathbf{W}_l \mathbf{x}_{l-1}$ is the pre-activation with weight matrix $\mathbf{W}_l$, $f$ is the activation function and $\mathbf{x}_l$ is the output from layer $l$. The input to our network is therefore denoted $\mathbf{x}_0$, and the output $\mathbf{x}_L$. We consider learning rules which update the weights of such a network of the form

$$\mathbf{W}_l \leftarrow \mathbf{W}_l - \eta \Delta \mathbf{W}_l$$

where $\eta$ is a small constant learning rate and $\Delta \mathbf{W}_l$ is a parameter update direction for a particular algorithm. Recall that the regular BP update is given by

$$\Delta \mathbf{W}_l = \mathbf{g}_l \mathbf{x}_{l-1}^\top \tag{1}$$

with $\mathbf{g}_l = \frac{\partial \mathcal{L}}{\partial \mathbf{a}_l}$ the gradient of the loss $\mathcal{L}$ with respect to the layer activations $\mathbf{a}_l$. Our aim is (1) to derive gradient approximations which have better properties than standard node perturbation and (2) to show that decorrelation massively improves convergence performance for any of the employed learning algorithms. In the following, we consider weight updates relative to a single input sample $\mathbf{x}_0$. In practice, these updates are averaged over mini-batches.

### 2.1.1 Iterative node perturbation

In the following, we develop a principled approach to node perturbation based learning. Our goal is to determine the gradient of the loss with respect to the pre-activations in a layer $l$ by use of perturbations. To this end, we consider the partial derivative of the loss with respect to the pre-activation $\mathbf{a}_l^i$ of unit $i$ in layer $l$ for all $i$. We define a perturbed state as $\tilde{\mathbf{x}}_k(h) = f(\tilde{\mathbf{a}}_k + h\mathbf{m}_k)$ with $h$ an arbitrary scalar and binary

vectors $\mathbf{m}_k = \mathbf{e}_i$ if $k = l$ with $\mathbf{e}_i$ a standard unit vector and $\mathbf{m}_k = \mathbf{0}$ otherwise. We may now define the partial derivatives as

$$(\mathbf{g}_l)_i = \lim_{h \to 0} \frac{\mathcal{L}(\tilde{\mathbf{x}}_L(h)) - \mathcal{L}(\mathbf{x}_L)}{h} \, .$$

This suggests that node perturbation can be rigorously implemented by measuring derivatives using perturbations $h\mathbf{m}_k$ for all units $i$ individually in each layer $l$. However, this would require as many forward-passes as there exist nodes in the network, which would be extremely inefficient.

An alternative approach is to define perturbations in terms of directional derivatives. Directional derivatives measure the derivative of a function based upon an arbitrary vector direction in its dependent variables. At this stage a choice can be made: either one may measure this directional derivative for a single layer or for an entire network. Measuring the directional derivative with respect to a specific layer via a perturbation given by

$$\tilde{\mathbf{x}}_k(h) = f\left(\tilde{\mathbf{a}}_k + h\mathbf{v}_k\right)$$

where $\mathbf{v}_k \sim \mathcal{N}(\mathbf{0}, \sigma^2 \mathbf{I}_k)$ if $k = l$ and $\mathbf{v}_k = \mathbf{0}$ otherwise. Here, $\sigma^2$ is a scalar, equivalent to the noise variance. Given this definition, we can precisely measure a directional derivative with respect to the activities of layer $l$ in our deep neural network, in vector direction $\mathbf{v} = (\mathbf{v}_1, \ldots, \mathbf{v}_L)$ as

$$\nabla_{\mathbf{v}} \mathcal{L} = \lim_{h \to 0} \frac{\mathcal{L}\left(\tilde{\mathbf{x}}_L(h)\right) - \mathcal{L}\left(\mathbf{x}_L\right)}{h\|\mathbf{v}\|}$$

where the directional derivative is measured by a difference in the loss induced in the vector direction $\mathbf{v}$, normalized by the vector length $\|\mathbf{v}\|$ and in the limit of infinitesimally small perturbation. The normalization ensures that the directional derivative is taken with respect to unit vectors. Note that this derivative is only being measured for layer $l$ as for all other layers the perturbation vector is composed of zeros.

As derived in Appendix A, by averaging the directional derivative across samples of the vector $\mathbf{v}$, in the limit $h \to 0$, we exactly recover the gradient of the loss with respect to a specific layer, such that

$$\mathbf{g}_l = N_l \left\langle \nabla_{\mathbf{v}} \mathcal{L} \frac{\mathbf{v}_l}{\|\mathbf{v}\|} \right\rangle_{\mathbf{v}} \, . \tag{2}$$

Equation 2 allows us to measure the gradient of a particular layer of a deep neural network by perturbation. Instead of averaging over multiple noise vectors, we use a sample-wise weight update and average over mini-batches instead. Fixing $h$ at a small value and incorporating it into the scale of the noise vector $\mathbf{v}$, we obtain weight update

$$\Delta \mathbf{W}_l^{\text{INP}} = N_l \, \delta \mathcal{L} \frac{\mathbf{v}_l}{\|\mathbf{v}\|^2} \, \mathbf{x}_{l-1}^{\top} \tag{3}$$

where we used $\nabla_{\mathbf{v}} \mathcal{L} = \delta \mathcal{L} / \|\mathbf{v}\|$. We refer to this method as iterative node perturbation (INP).

### 2.1.2 Node perturbation

Further zooming out, one may wish to perturb all nodes in an entire network, instead of just one layer at a time. In regular node perturbation, noise is injected into all layer's pre-activations and weights are updated in the direction of the noise if the loss improves and in the opposite direction if it worsens. As with iterative node perturbation, two forward passes are required: one clean and one noise-perturbed. During the noisy pass, noise is injected into the pre-activation of each layer to yield a perturbed output

$$\tilde{\mathbf{x}}_l = f\left(\tilde{\mathbf{a}}_l + \boldsymbol{\epsilon}_l\right) = f\left(\mathbf{W}_l \tilde{\mathbf{x}}_{l-1} + \boldsymbol{\epsilon}_l\right) \tag{4}$$

where the added noise $\boldsymbol{\epsilon}_l \sim \mathcal{N}(\mathbf{0}, \sigma^2 \mathbf{I}_l)$ is a spherical Gaussian perturbation with no cross-correlation and $\mathbf{I}_l$ is an $N_l \times N_l$ identity matrix with $N_l$ the number of nodes in layer $l$. Note that this perturbation has a cumulative effect on the network output as each layer's perturbed output $\tilde{\mathbf{x}}_l$ is propagated forward through the network, resulting in layers deeper in the network being perturbed by more than just their own added noise.

As before, one can measure a loss differential for an NP-based update. Supposing that the loss $\mathcal{L}$ is measured using the network outputs, the loss difference between the clean and noisy network is given by $\delta\mathcal{L} = \mathcal{L}(\tilde{\mathbf{x}}_L) - \mathcal{L}(\mathbf{x}_L)$, where $\delta\mathcal{L}$ is a scalar measure of the difference in loss induced by the addition of noise to the network. Given this loss difference and the network's perturbed and unperturbed outputs, we compute a layer-wise learning signal by replacing the gradient $\mathbf{g}_l$ in Eq. 1 with a term consisting of the loss difference and a normalized noise vector:

$$\Delta \mathbf{W}_l^{\text{NP}} = \delta\mathcal{L}\frac{\boldsymbol{\epsilon}_l}{\sigma^2}\mathbf{x}_{l-1}^\top \tag{5}$$

This update is similar to that of INP, with one key difference, that the normalization term, $N_l/||\mathbf{v}||^2$, is approximated as the inverse of the node-wise variance, $1/\sigma^2$. Thus, regular NP is a network-wide measure of the directional derivative with an approximation of the required normalization.

NP thus explores in a larger parameter space than INP. Importantly, however, it requires knowledge of each layer's noise vector, $\epsilon_l$. This knowledge is hard to attain in real-world cases of noise injection within a neural network as this noise has to be distinguishable from the noise induced by the previous layers.

### 2.1.3 Activity-based node perturbation

To improve upon NP and to mitigating the need for exact and separate measurement of the noise at every layer, we propose to approximate the directional derivative across the whole network simultaneously based upon the activity-differences. This can be achieved by, instead of measuring and tracking the injected noise alone, measuring the state difference between the clean and noisy forward passes of the network. Concretely, taking the definition of the forward pass given by NP in Eq. 4, we define

$$\Delta \mathbf{W}_l^{\text{ANP}} = N\,\delta\mathcal{L}\frac{\delta\mathbf{a}_l}{||\delta\mathbf{a}||^2}\,\mathbf{x}_{l-1}^\top \tag{6}$$

where $N = \sum_{l=0}^{L} N_l$ is the number of units in the network, $\delta\mathbf{a}_l = \tilde{\mathbf{a}}_l - \mathbf{a}_l$ is the activity difference between a noisy and a clean pass in layer $l$ and $\delta\mathbf{a} = (\delta\mathbf{a}_1, \ldots, \delta\mathbf{a}_L)$ is the concatenation of activity differences. The resulting update rule is referred to as activity-based node perturbation (ANP). Appendix B provides a derivation of this rule.

Here, we have now updated multiple aspects of the NP rule. First, rather than using the measure of isolated noise injected at each layer, we instead measure the change in activation between the clean and noisy networks. This has a theoretical benefit as well as a theoretical downside. As a theoretical benefit, by using the total noise at any given layer, this subsumes noise in all past layers and allows one to update a layer as if there was zero noise added to the preceeding layers. The only correlative noise issues remaining are then with the later network layers. ANP is thus demonstrably better than NP for layers closer to the output of a network.

On the other hand, the noise at each node in a layer is no longer independent and could now have a correlated structure. This is a downside in that it can skew the gradient descent direction, though can be shown to still converge to a minimum in loss. Both of these implications are expanded upon in Appendix B. Second, this direct use of the activity difference also requires a recomputation of the scale of the perturbation vector and the requisite normalization. Unlike in the case of NP, one cannot assume the size of this normalization.

Note that the ANP rule is now a hybridization of iterative node perturbation and traditional node perturbation. In the limit where the nodes are not interdependent (i.e. when we have a single layer), ANP is equivalent to INP. When used in a layered network structure, ANP allows computation of updates without explicit access to the noise vectors and updates an entire network in the same style as traditional NP.

## 2.2 Increasing perturbation learning efficiency through decorrelation

Uncorrelated data variables have been proposed and demonstrated as impactful in making credit assignment more efficient in deep neural networks (LeCun et al., 2002). If a layer's inputs, $\mathbf{x}_l$, have highly correlated features, a change in one feature can be associated with a change in another correlated feature, making it difficult for the network to disentangle the contributions of each feature to the loss. This can lead to less efficient learning, as has been described in previous research in the context of BP (Luo, 2017; Wadia et al.,

2021). NP accordingly benefits from decorrelation of input variables at every layer. Specifically, Hiratani et al. (2022) demonstrate that the covariance of NP updates between layers $k$ and $l$ can be described as

$$C_{kl}^{\text{np}} \approx 2C_{kl}^{\text{sgd}} + \delta_{kl} \left\langle \sum_{m=1}^{k} \|\mathbf{g}_m\|^2 \mathbf{I}_k \otimes \mathbf{x}_{k-1}\mathbf{x}_{k-1}^T \right\rangle_{\mathbf{x}} \tag{7}$$

where $C_{kl}^{\text{sgd}}$ is the covariance of SGD updates, $\delta_{kl}$ is the Kronecker delta and $\otimes$ is a tensor product. Equation 7 implies that in NP the update covariance is twice that of the SGD updates plus an additional term that depends on the correlations in the input data $\mathbf{x}_{k-1}\mathbf{x}_{k-1}^T$. Removing correlations from the input data should therefore reduce the bias in the NP algorithm updates, possibly leading to better performance.

In this work, we introduce decorrelated node perturbation, in which we decorrelate each layer's input activities using a trainable decorrelation procedure first described by Ahmad et al. (2023). A layer input $\mathbf{x}_l$ is decorrelated by multiplication by a decorrelation matrix $\mathbf{R}_l$ to yield a decorrelated input $\mathbf{x}_l^\star = \mathbf{R}_l\mathbf{x}_l$. The decorrelation matrix $\mathbf{R}_l$ is then updated according to

$$\mathbf{R}_l \leftarrow \mathbf{R}_l - \alpha \left( \mathbf{x}_l^\star (\mathbf{x}_l^\star)^\top - \text{diag}\left( (\mathbf{x}_l^\star)^2 \right) \right) \mathbf{R}_l$$

where $\alpha$ is a small constant learning rate and $\mathbf{R}_l$ is initialized as the identity matrix. For a full derivation of this procedure see (Ahmad et al., 2023).

Decorrelation effectively aligns the gradient updates more closely with those of the natural gradient (Ahmad, 2024) and can be combined with any of the formulations described in the previous sections by replacing $\mathbf{x}_l$ in Eq. 1 with $\mathbf{x}_l^*$. We will use DBP, DNP, DINP and DANP when referring to the decorrelated versions of the described learning rules. Algorithm 1, describes the decorrelated activity-based node perturbation procedure, which we will argue is of particular interest for implementations on unconventional computing hardware.

---

**Algorithm 1** Decorrelated activity-based node perturbation (DANP)

---

**Input:** *data $\mathcal{D}$, network $\{(\mathbf{W}_l, \mathbf{R}_l)\}_{l=1}^L$, learning rates $\eta$ and $\alpha$*
**for each** *epoch* **do**
    **for each** $(\mathbf{x}_0, \mathbf{t}) \in \mathcal{D}$ **do**
        **for** layer $l$ **from** 1 **to** $L$ **do**                                        ▷ Regular forward pass
            $\mathbf{x}_{l-1}^\star = \mathbf{R}_{l-1}\mathbf{x}_{l-1}$
            $\mathbf{a}_l = \mathbf{W}_l\mathbf{x}_{l-1}^\star$
            $\mathbf{x}_l = f(\mathbf{a}_l)$
        **end for**
        $\tilde{\mathbf{x}}_0 = \mathbf{x}_0$
        **for** layer $l$ **from** 1 **to** $L$ **do**                                          ▷ Noisy forward pass
            $\tilde{\mathbf{x}}_{l-1}^\star = \mathbf{R}_{l-1}\tilde{\mathbf{x}}_{l-1}$
            $\tilde{\mathbf{a}}_l = \mathbf{W}_l\tilde{\mathbf{x}}_{l-1}^\star + \boldsymbol{\epsilon}_l$
            $\tilde{\mathbf{x}}_l = f(\tilde{\mathbf{a}}_l)$
        **end for**
        $\delta\mathcal{L} = (\mathbf{t} - \tilde{\mathbf{x}}_L)^2 - (\mathbf{t} - \mathbf{x}_L)^2$                            ▷ Compute loss difference
        **for** layer $l$ **from** 1 **to** $L$ **do**
            $\mathbf{W}_l \leftarrow \mathbf{W}_l - \eta N \,\delta\mathcal{L} \frac{\tilde{\mathbf{a}}_l - \mathbf{a}_l}{||\delta\mathbf{a}||^2} \left( \mathbf{x}_{l-1}^\star \right)^\top$             ▷ Update weight matrix
            $\mathbf{R}_l \leftarrow \mathbf{R}_l - \alpha \left( \mathbf{x}_l^\star (\mathbf{x}_l^\star)^\top - \text{diag}\left( (\mathbf{x}_l^\star)^2 \right) \right) \mathbf{R}_l$     ▷ Update decorrelation matrix
        **end for**
    **end for**
**end for**

---

### 2.3 Experimental validation

Our experiment in Section 3.1 uses a synthetic input–output mapping, where input vectors $\mathbf{x}$ are sampled from a Gaussian distribution $\mathcal{N}(\mathbf{0}, \mathbf{1})$. When computing updates, perturbations are applied by adding small random noise to layer activations, where each perturbation vector is drawn from $\mathcal{N}(\mathbf{0}, \sigma^2)$ with $\sigma^2 = 10^{-6}$. These

updates are then compared to updates computed by BP to determine their correspondence by measuring the angle, measured as $\cos\theta = \frac{\mathbf{u}\cdot\mathbf{v}}{|\mathbf{u}||\mathbf{v}|}$, where $\theta$ is the angle, $\mathbf{u}$ and $\mathbf{v}$ are the vectors, and $|\cdot|$ denotes the Euclidean norm. For all other experiments in this work using NP, ANP or INP $\sigma^2_{\text{pert}} = 10^{-6}$ was also used when sampling the perturbations. See Appendix C for more details on how this value was determined.

To measure the performance of the algorithms proposed, we ran a set of experiments with the CIFAR-10 (Krizhevsky, 2009) and Tiny ImageNet datasets (Le and Yang, 2015), using fully-connected and convolutional neural networks, specifically aiming to quantify the performance differences between the traditional (NP), layer-wise iterative (INP) and activity-based (ANP) formulations of NP as well as their decorrelated counterparts. These datasets were chosen as they have previously been shown to yield low performance when training networks using regular node perturbation, demonstrating its limitations (Hiratani et al., 2022). All experiments were repeated using three random seeds, after which performance statistics were aggregated. Further experimental details can be found in Appendix D. Additionally, we performed experiments on the SARCOS dataset (Vijayakumar and Schaal, 2000), demonstrating the application of our work in a robotics setting.

## 3 Results

### 3.1 INP and ANP align better with the true gradient

In single-layer networks, the three described NP formulations (NP, INP and ANP) converge to an equivalent learning rule. Therefore, multi-layer networks with three hidden layers were used to investigate performance differences across our proposed formulations.

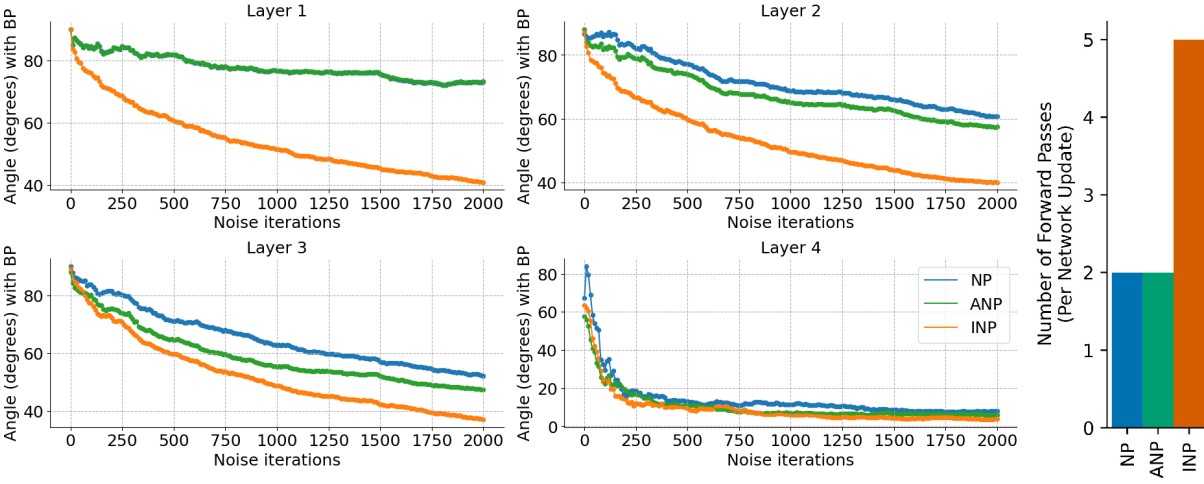

Figure 1: Left: Angles between BP's weight update and an update calculated by NP, ANP and INP as a function of the number of noise iterations. Note that the network is not updated across the 'Noise iterations' dimension, but instead multiple noise samples are propagated through a fixed network and their updates averaged. This is computed for a fully-connected, 3 hidden layer, network with leaky ReLU nonlinearities. Right: Number of forward passes for NP, ANP and INP.

Figure 1, left, shows how weight updates of the different node perturbation methods compare to those from BP in a three-hidden-layer, fully-connected, feedforward network trained on synthetic data. Specifically, it is shown how each of these methods compare in the angle of their update relative to BP when you average their update over a greater and greater number of noise samples. Note that the update for Layer 1 is identical for NP and ANP.

When measuring the angles between the update vectors of various methods, we can observe that the INP method is by far the most well-aligned in its updates with respect to backpropagation, followed by ANP and closely thereafter by NP. These results align with the theory laid out in the methods section of this work. Appendix E shows the same data as a function of the number of noisy forward passes.

Note that for all of these algorithms, alignment with BP updates improves when updates are averaged over more samples of noise. Therefore, it is the ranking of the angles between the NP algorithms that is of interest here, not the absolute value of the angle itself, as all angles would improve with more noise samples. Appendix C shows that our results do not depend strongly on the noise variance $\sigma^2$, producing similar results when varying $\sigma^2$ by a few orders of magnitude.

Figure 1, right, shows that the node perturbation methods also differ in how many forward passes are required. Specifically, although all algorithms use the same dimensionality of noise the INP method requires an individual forward pass for each layer to isolate the impact of this noise and to better approach the BP update. In the following we compare INP to the other node perturbation variants without accounting for this excess number of forward-passes, but a reader should bear in mind this drawback and consider the added computational complexity of INP.

## 3.2 Decorrelation improves convergence

To assess the impact of decorrelation on NP's performance, we studied a single-layer network trained with NP, DNP, BP and DBP. Note that the different formulations of NP are identical in a single-layer networks where there is no impact on activities in the layer from previous layers.

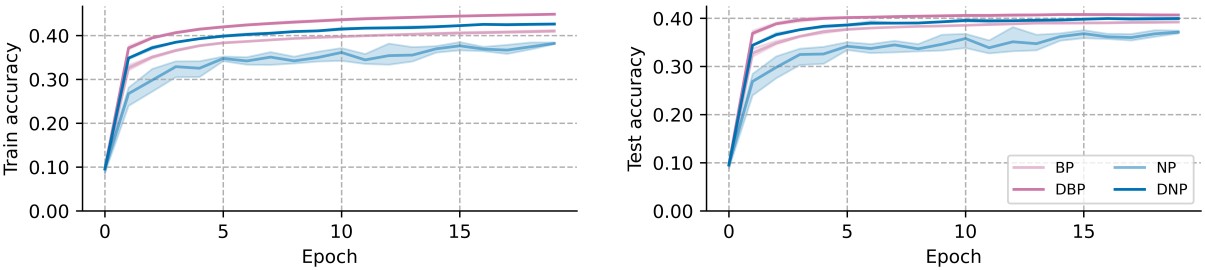

Figure 2: Performance of (D)NP and (D)BP on CIFAR-10 when training a single-layer architecture.

Figure 2 indicates that, in a single-layer context, DNP outperforms NP significantly and performs slightly better than BP, with DBP performing better still. It appears that part of the benefit of decorrelation is due to a lack of correlation in the input unit features and a corresponding ease in credit assignment without confound. An additional benefit from decorrelation, that is specific to NP-style updates, is explained by the way in which decorrelation reduces the covariance of NP weight updates, as described in the methods section above.

## 3.3 Multi-layer networks can be trained effectively with node perturbation

We proceed by determining the performance of the different algorithms in multi-layer neural networks. See Appendix F, Table 3 for peak accuracies. Figure 3 shows the performance when training a multi-layer fully connected network. We see all decorrelated algorithms convincingly outperforming their non-decorrelated counterparts. We also see (D)INP outperform (D)ANP and (D)NP, almost keeping up with BP. Performance benefits of (D)ANP over (D)NP are not as clear in these results. Note that DNP performs much better in a three-hidden-layer network than in the single-layer networks, indicating that DNP does facilitate multi-layer credit assignment much more than regular NP, which actually performs worse in the deeper network.

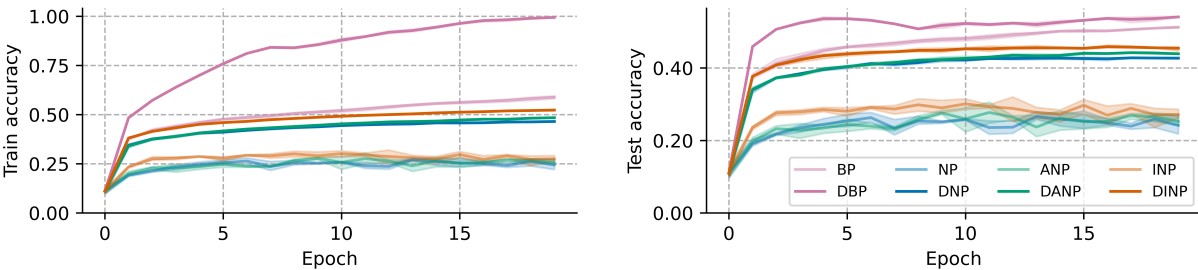

Figure 3: Performance of (D)NP, (D)ANP, (D)INP and (D)BP on CIFAR-10 in a three-hidden layer network. For both figures, curves report mean train and test accuracy. Shaded areas indicate the maximal and minimal accuracy obtained for three random seeds. Note that all NP methods have an equivalent formulation in a single-layer network.

## 3.4 Resampling allows for scaling to larger systems

To further investigate the capabilities of our approach, we apply BP, DNP, DANP and DINP to the more challenging Tiny ImageNet dataset (Le and Yang, 2015) using a deeper architecture consisting of three convolutional layers and three fully connected layers. The Tiny ImageNet dataset has 200 classes and thus has a larger output space than CIFAR-10. Larger output spaces are known to be challenging for traditional NP-style algorithms (Hiratani et al., 2022). RGB-values for the images, which are ImageNet images downsized to $64 \times 64$ pixels, were standardized as a preprocessing step. For network architecture and learning rates, see Appendix D.

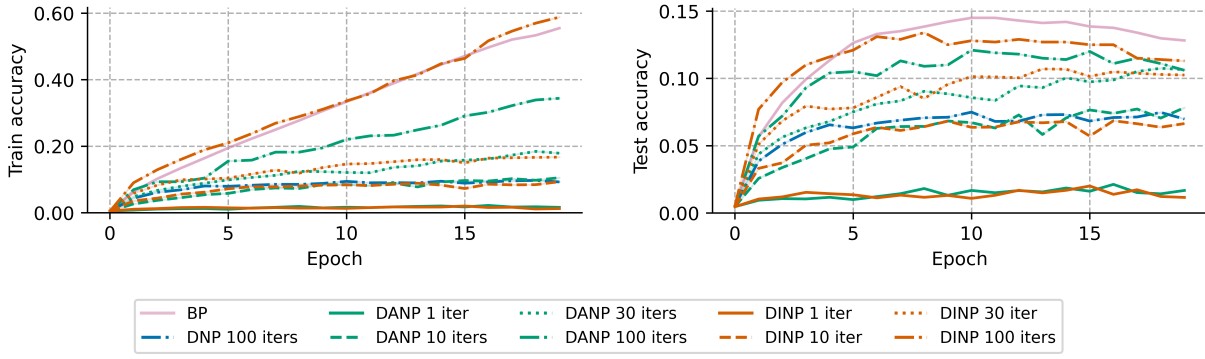

Figure 4: Train and test accuracy for BP, DNP, DANP and DINP on Tiny ImageNet. DANP and DINP applied with 1, 10, 30 and 100 noise samples per update step. DNP applied with 100 noise samples per update step.

Figure 4 shows that, in their naive implementation, DNP, DANP and DINP do not appreciably learn this task. We demonstrate that, for such a difficult task, this is due to the accuracy of the gradient estimation. To show this, we resampled the random noise 10, 30 and 100 times before applying each update, effectively carrying out 10/30/100 noisy passes to compare against each single clean pass. For DNP, only an experiment with 100 iterations was included, as this algorithm did not match the performance of DINP and DANP. Results again demonstrate that both DANP and DINP improve meaningfully on DNP in deeper networks. The additional noise sampling comes at significant computational expense but could in principle be parallelized with sufficiently powerful hardware. The results indeed show that, as our theory suggests, sampling the noise distribution multiple times per update step significantly improves alignment with backpropagation, with DINP with 100 samples per update closely approximating BP's learning trajectory. See Appendix F, Table 4 for peak accuracies.

### 3.5 Robot control task

As an additional demonstration that our results generalize to other datasets with applications in robotics, we report results on the SARCOS dataset (Vijayakumar and Schaal, 2000).

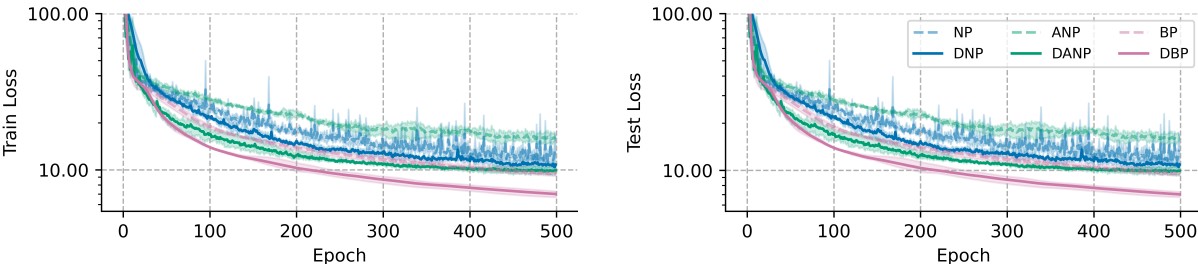

Figure 5: Train and test loss for (D)NP, (D)ANP and (D)BP on the SARCOS dataset. Curves report mean train and test loss. Error bars indicate the maximal and minimal loss obtained for three random seeds.

Figure 5 shows train and test loss for (D)NP, (D)ANP and (D)BP for the SARCOS dataset (Vijayakumar and Schaal, 2000). The task is to solve inverse dynamics problem for a seven degrees-of-freedom SARCOS anthropomorphic robot arm by mapping from a 21-dimensional input space (7 joint positions, 7 joint velocities, 7 joint accelerations) to the corresponding 7 joint torques. The network is a fully connected architecture with two 32-unit hidden layers and was trained with MSE loss and the leaky ReLU activation function. The network was trained for 500 epochs and results were averaged over three random seeds. A learning rate search for each algorithm was started at $10^{-7}$, doubling the learning rate until the algorithm became unstable. A decorrelation learning rate of $10^{-4}$ was used for DNP and DANP and $10^{-3}$ for DBP.

The results indicate that while NP and ANP lag BP in performance, DNP and DANP perform almost as well as BP. Like in other experiments reported in this work, DBP performs best overall. Final test losses were 11.18 (NP), 10.51 (DNP), 15.86 (ANP), 9.79 (DANP), 9.48 (BP) and 7.02 (DBP).

### 3.6 Node perturbation allows learning in noisy systems

One of the most interesting applications of perturbation-based methods for learning are for systems that are inherently noisy but have the property that the noise cannot be measured directly. This includes both biological nervous systems as well as a range of analog computing and neuromorphic hardware architectures (Kaspar et al., 2021).

To demonstrate that our method is also applicable to architectures with embedded noise, we train networks in which there is no clean network pass available. Instead, two noisy network passes are computed and one is taken as if it were the clean pass. That is, we use

$$\delta\mathbf{a}_l = \tilde{\mathbf{a}}_l^{(1)} - \tilde{\mathbf{a}}_l^{(2)}$$

in Eq. 6, where both $\tilde{\mathbf{a}}_l^{(1)}$ and $\tilde{\mathbf{a}}_l^{(2)}$ are generated by running a forward pass under noise perturbations. This is similar in spirit to the approach suggested by Cho et al. (2011). In this case, we specifically compare DANP to DNP in terms of their robustness to a noisy baseline. DANP does not assume that the learning algorithm can independently measure noise. Instead it can only measure the present, and potentially noisy, activity and thereafter measure activity differences to guide learning.

Figure 6 shows that computing updates based on a set of two noisy passes, rather than a clean and noisy pass, produces extremely similar learning dynamics for DANP with some minimal loss in performance. DNP, in contrast, learns more slowly during early training and becomes unstable late in training, decreasing in train accuracy. It also does not reach the same level of test performance as it does without the noisy baseline. The similarity of the speed of learning and performance levels for DANP suggests that clean network passes may provide little additional benefit for the computation of updates in this regime. These results are most

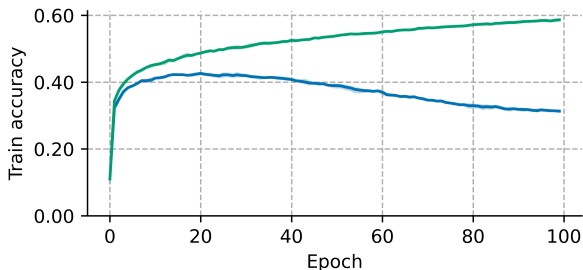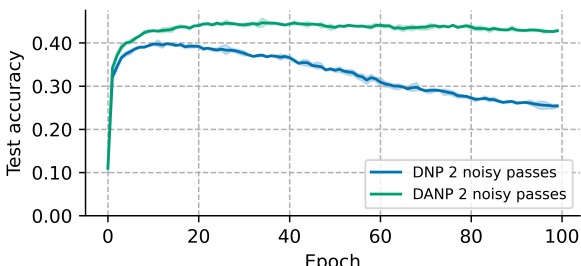

Figure 6: Performance of DNP and DANP when using two noisy network passes. Results are reported for a three-layer fully-connected network trained for CIFAR-10 classification (comparable to Figure **??**, bottom panel). Curves report mean train and test accuracy. Shaded areas indicate the maximal and minimal accuracy obtained for three random seeds.

promising for application of DANP to systems in which noise is an inherent property and cannot be selectively switched off or read out.

## 4   Discussion

In this work, we explored several formulations of NP learning and also introduced a layer-wise decorrelation method which strongly outperforms baseline implementations. In our results we show robust speedups in training compared to regular NP, suggesting that our alternative formulations of NP could prove to be competitive with traditional BP in certain contexts. We attribute this increased efficacy to a better alignment with theoretically derived directional derivatives and a more efficient credit assignment by virtue of decorrelated input features at every layer.

The performance of all of our proposed methods is substantially improved by including decorrelation. When comparing NP, INP and ANP, a number of observations are in order. First, INP yields the best results stemming from its robust relationship to the directional derivative and underlying gradient. The INP method does, however, require running as many forward passes as there are layers in the network (plus one 'clean' pass). This is a drawback for this method and should be considered when analyzing its performance, though it should also be considered that it could remain efficient on hardware in which all forward passes are parallelizable. Second, ANP consistently outperforms NP. This can be attributed to its ability to subsume all noise from previous layers into a given layer's update and to not have any issues of correlation with these past layers. Finally, decorrelation aids all NP variants to achieve higher performance and faster convergence.

It is important to note that adding decorrelation matrices to a neural network does increase the amount of computation required in the forward pass. Appendix G provides details on the exact amount of additional computation and memory required when running BP, DBP or DANP. It should be noted, however, that several computational tricks are available that will significantly reduce said computational burden. These methods have not been applied in the current work, due to the small architectures trained, but would likely benefit runtimes in larger networks. First, the decorrelation matrix $\mathbf{R}$ can be fused with forward weights $\mathbf{W}$, such that $\mathbf{A} = \mathbf{WR}$ and $\mathbf{x} = \mathbf{Az}$, moving the multiplication by $\mathbf{R}$ to $\mathbf{W}$, which does not contain a batch dimension, significantly reducing computation in the forward pass. Second, the update of $\mathbf{R}$ need not be estimated using the entire mini-batch, but can also be estimated using a heavily downsampled mini-batch (Dalm et al., 2024). Third, in the case of convolutional layers, only the local image patch on which the convolutional kernel operates could be decorrelated (Dalm et al., 2024). As the only correlations that matter when updating a convolutional kernel are the local correlations within its receptive field, this should not diminish performance gains, while greatly reducing computational overhead.

Testing the scalability of these approaches to tasks of greater complexity is also important, as are their application to other network architectures such as residual networks, recurrent networks and attention-based models. Training accuracy, of even the best NP variants, also show some lag behind backpropagation's

accuracy achievements in deeper networks. Further investigation is needed to determine how to bridge this gap and if it has implications for harder tasks. Our scaling results do demonstrate that by resampling the noise we achieve better and better alignment with the gradient direction, by virtue of the formal correspondence between directional derivatives and our formulations of node perturbation. In this sense, our approach can be viewed as an alternative to forward accumulation, where we exchange memory complexity (maintaining all partial derivatives) for time complexity (repeated forward passes with resampled noise vectors).

Noise-based learning approaches might also be ideally suited for implementation on neuromorphic hardware (Kaspar et al., 2021). First, as demonstrated in Figure 6, our proposed DANP algorithm scales well even when there is no access to a 'clean' model pass. Furthermore, DANP does not require access to the noise signal itself, in contrast to NP. This means that such an approach could be ideally suited for implementation in noisy physical devices (Gokmen, 2021) even in case the noise cannot be measured. Even on traditional hardware architectures, forward passes are often easier to optimize than backward passes and are often significantly faster to compute. This can be especially true for neuromorphic computing approaches, where backward passes require automatic differentiation implementations and a separate computational pipeline for backward passes (Zenke and Neftci, 2021). In these cases, a noise-based approach to learning could prove highly efficient.

Exploring more efficient forms of noise-based learning is interesting beyond credit-assignment alone. This form of learning is more biologically plausible as it does not require weight transport nor any specific feedback processing (Grossberg, 1987; Crick, 1989; Lillicrap et al., 2016). There is ample evidence for noise in biological neural networks (Faisal et al., 2008) and we suggest here that this could be effectively used for learning. Furthermore, the positive impact of decorrelation on learning warrants further investigation of how this mechanism might be involved in neural plasticity. It is interesting to note that various mechanisms act to reduce correlation or induce whitening, especially in early visual cortical areas (King et al., 2013). Additionally, lateral inhibition, which is known to occur in the brain, can be interpreted as a way to reduce redundancy in input signals akin to decorrelation, making outputs of neurons less similar to each other (Békésy, 1967). As described in (Ahmad et al., 2023), decorrelation updates can rely exclusively on information that is locally available to the neuron, making it amenable to implementation in biological or physical systems.

In general, our work opens up exciting opportunities since it has the potential to bring gradient-free training of deep neural networks within reach. That is, in addition to not requiring a backward pass, efficient noise-based learning may also lend itself to networks not easily trained by backpropagation, such as those consisting of activation functions with jumps, binary networks or networks in which the computational graph is broken, as in reinforcement learning.

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

## A   The directional derivative as a measure of the gradient

In the main text, we relate the measurement of directional derivatives to the gradient. Specifically, for a feedforward deep neural network of $L$ layers, we state that

$$\mathbf{g}_l = N_l \left\langle \nabla_{\mathbf{v}} \mathcal{L} \frac{\mathbf{v}_l}{||\mathbf{v}||} \right\rangle_{\mathbf{v}}$$

where $\mathbf{v} = (\mathbf{v}_1, \ldots, \mathbf{v}_L)$ and $\mathbf{v}_k \sim \mathcal{N}(\mathbf{0}, \sigma^2 \mathbf{I}_k)$ if $k = l$ and $\mathbf{v}_k = \mathbf{0}$ otherwise.

Let us now demonstrate the equivalence of the gradient to the expectation over this directional derivative. For simplicity, let us consider a single layer network, such that $L = 1$ and $\mathbf{v} = \mathbf{v}_l = \mathbf{v}_1$. A directional derivative can always be equivalently written as the gradient vector dot product with a (unit-length) direction vector, such that

$$\nabla_{\mathbf{v}} \mathcal{L} = \nabla \mathcal{L}^\top \frac{\mathbf{v}}{||\mathbf{v}||} \,.$$

Substituting this form into the above equation requires ensuring that our directional derivative term is no longer treated as a scalar but as a $\mathbb{R}^{1 \times 1}$ matrix. We also a transpose the gradient vector and then untranspose the entire expression to allow derivation with clarity. Further, we remove the now redundant subscripts, such that:

$$\mathbf{g} = N \left\langle \nabla_{\mathbf{v}} \mathcal{L} \frac{\mathbf{v}^\top}{||\mathbf{v}||} \right\rangle_{\mathbf{v}}^\top = N \left\langle \nabla \mathcal{L}^\top \frac{\mathbf{v}}{||\mathbf{v}||} \frac{\mathbf{v}^\top}{||\mathbf{v}||} \right\rangle_{\mathbf{v}}^\top = N \left\langle \frac{\mathbf{v}\mathbf{v}^\top}{||\mathbf{v}||^2} \right\rangle_{\mathbf{v}}^\top \nabla \mathcal{L} \,.$$

If we assume that our noise distribution is composed of independent noise with a given variance, $\mathbf{\Sigma} = \sigma^2 \mathbf{I}$ then we obtain

$$\mathbf{g} = N \left\langle \frac{\mathbf{v}\mathbf{v}^\top}{||\mathbf{v}||^2} \right\rangle_{\mathbf{v}}^\top \nabla \mathcal{L} \propto N \frac{\mathbf{I}}{N} \nabla \mathcal{L} = \nabla \mathcal{L} \,.$$

Note that normalization of every vector means that the correlations $\left\langle \mathbf{v}\mathbf{v}^\top / ||\mathbf{v}||^2 \right\rangle$ are not changed in sign but only in scale, as the noise vectors are now all scaled to lie on a unit sphere (without any rotation). In our specific case of a diagonal correlation matrix, this expectation is also therefore diagonal and all diagonal elements are equal. The average value of these diagonal elements is then simply the variance $1/N$ of a random unit vector in $N$ dimensional space. For other noise distributions, in which there does exist cross-correlation between the noise elements, this is no longer an appropriate treatment. Instead, one would have to multiply by the inverse of the correlation matrix to fully recover the gradient values.

## B   Derivation of activity-based node perturbation

Here we explicitly demonstrate how the INP rule can give rise to the ANP rule under some limited assumptions. Consider the INP learning rule, which is designed to return a weight update for a specific layer of a DNN,

$$\Delta \mathbf{W}_l^{\text{INP}} = N_l \, \delta \mathcal{L} \frac{\mathbf{v}_l}{||\mathbf{v}||^2} \, \mathbf{x}_{l-1}^\top \,.$$

This rule has been derived in the main text in order to optimally make use of noise to determine the directional derivative with respect to a single layer of a deep network, $l$, while all other layers receive no noise. For this purpose, the noise vector is defined such that $\mathbf{v}_k \sim \mathcal{N}(\mathbf{0}, \sigma^2 \mathbf{I}_k)$ if $k = l$ and $\mathbf{v}_k = \mathbf{0}$ otherwise. In moving from INP to ANP, we aim to fulfill the following conditions. First, update a whole network in one pass rather than per layer. Second, allow updating without explicit access to the noise at every layer, but instead access only to the activity difference.

These goals can be accomplished in a simple manner. To achieve the first goal, we can simply treat the whole network as if it is a single layer (even if it is not). The only change required for this modification is to assume that this rule holds even if $\mathbf{v}_k \sim \mathcal{N}(\mathbf{0}, \sigma^2 \mathbf{I}_k)$ for $k \in [1, \ldots, L]$. I.e., that noise is injected for all layers and all

layers are simultaneously updated. To achieve the second goal, we can assume that one does not have access to the noise vector directly, but instead has access to the output activations from a clean and noisy pass, $\mathbf{a}_l$ and $\tilde{\mathbf{a}}_l$ respectively. Thus, rather than measuring the noise directly, one can measure the impact of all noise upon the network by substituting the noise vector, $\mathbf{v}$ for the activity difference $\delta\mathbf{a} = \tilde{\mathbf{a}} - \mathbf{a}$. Thus, with these two modifications we arrive at the ANP learning rule

$$\Delta\mathbf{W}_l^{\mathrm{ANP}} = N\,\delta\mathcal{L}\frac{\delta\mathbf{a}_l}{||\delta\mathbf{a}||^2}\,\mathbf{x}_{l-1}^{\top}$$

which can be computed via two forward passes only.

## C   Impact of perturbation scale on performance

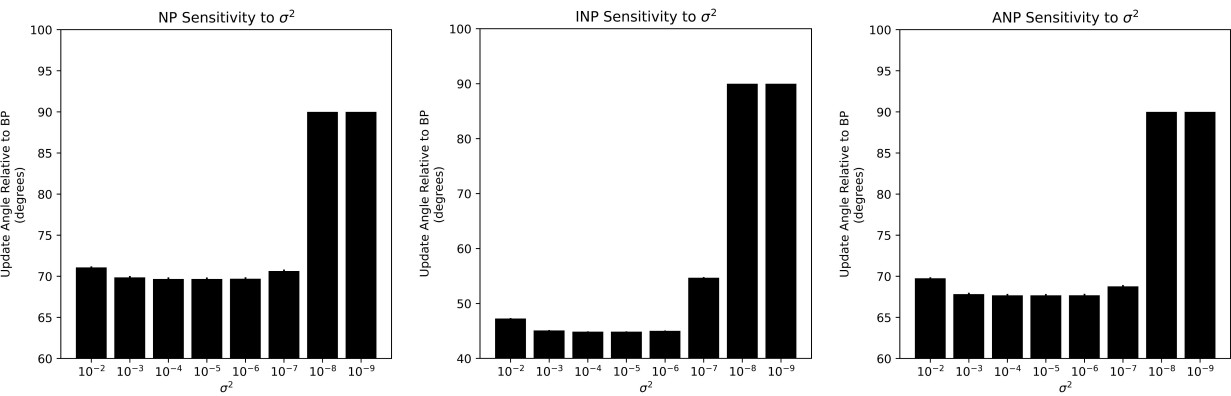

Figure 7: The alignment of NP, INP, and ANP updates measured against BP are shown across a range of perturbation parameters ($\sigma^2$). These angles are measure from updates computed for the second hidden layer of a randomly initialized three-hidden layer network being using a 1000 sample mini-batch from the CIFAR-10 dataset. The network from which these are sampled is equivalent to that used in Figure 1.

In all simulations of this paper, random noise is drawn from a set of independent Gaussian distributions with zero mean and some variance, $\sigma^2$. As can be seen in Figure 7, the NP, INP, and ANP methods show a robust performance (when measured based upon alignment with BP updates) to the choice of this variance of the perturbation distribution. For values across multiple orders of magnitude, from $10^{-4}$ to $10^{-6}$, there performance is completely stable. For smaller values of the variance, $\sigma^2 < 10^{-7}$, precision errors cause a decrease in performance (increase in angle). And for larger values of the variance, $\sigma^2 > 10^{-3}$, the size of the perturbation begins to affect network dynamics and thus the approximation of the directional derivative. In this work we exclusively use $\sigma^2 = 10^{-6}$, the smallest value possible before precision errors become an issue. We do not find significantly different results, even if $\sigma^2$ is set to be an order of magnitude greater.

## D   Experimental details

Experiments were performed using fully-connected or convolutional neural networks with leaky ReLU activation functions and a categorical cross-entropy (CCE) loss on the one-hot encoding of class membership. The Adam optimizer with default parameters $\beta_1 = 0.9$, $\beta_2 = 0.999$, $\epsilon = 1e-7$, was used for optimization. All experiments were run on an HP OMEN GT13-0695nd desktop computer with an NVIDIA RTX 3090 GPU. The experiments required about a week of total computation time and several more weeks of computation time during exploratory experiments. Table 1 provides details of the employed neural network architectures.

Table 1: The neural network architectures consist of fully connected (FC) and convolutional (Conv) layers. All layers except the output layer are followed by a leaky ReLU transformation. Convolutional layer size is given as height×width×output channels, stride.

| Network | Layer Types | Layer size |
|---|---|---|
| Single layer | FC | 10 |
| Three hidden layers | 3 × FC | 1024 |
| | FC | 10 |
| Six hidden layers | 6 × FC | 1024 |
| | FC | 10 |
| Nine hidden layers | 9 × FC | 1024 |
| | FC | 10 |
| ConvNet | Conv | 3×3×16, 2 |
| | Conv | 3×3×32, 2 |
| | Conv | 3×3×64, 1 |
| | FC | 1024 |
| | FC | 10 |
| ConvNet Tiny ImageNet | Conv | 5×5×32, 2 |
| | Conv | 3×3×64, 2 |
| | Conv | 5×5×128, 1 |
| | FC | 512 |
| | FC | 512 |
| | FC | 200 |

Table 2 shows the learning rates $\eta$ used in each experiment. For each experiment, the highest stable learning rate was selected. For the decorrelation learning rate a fixed value of $\alpha = 10^{-3}$ was chosen based on a prior manual exploration. Note that the minibatch size was fixed for every method at 1000. When using NP methods, activity perturbations were drawn from a univariate Gaussian with variance $\sigma^2 = 10^{-6}$.

Table 2: Learning rates used for different learning algorithms and architectures.

| Method | 1 layer | 3 hidden layers | 6 hidden layers | 9 hidden layers | ConvNet | ConvNet Tiny ImageNet |
|---|---|---|---|---|---|---|
| NP | $1.0 \times 10^{-3}$ | $1.0 \times 10^{-4}$ | – | – | – | – |
| DNP | $1.0 \times 10^{-3}$ | $1.0 \times 10^{-3}$ | $5.0 \times 10^{-3}$ | $5.0 \times 10^{-3}$ | – | $1.0 \times 10^{-3}$ |
| ANP | – | $1.0 \times 10^{-4}$ | – | – | $1.0 \times 10^{-4}$ | – |
| DANP | – | $1.0 \times 10^{-3}$ | $5.0 \times 10^{-3}$ | $5.0 \times 10^{-3}$ | $1.0 \times 10^{-4}$ | $1.0 \times 10^{-3}$ |
| INP | – | $1.0 \times 10^{-4}$ | – | – | $1.0 \times 10^{-4}$ | – |
| DINP | – | $1.0 \times 10^{-3}$ | $5.0 \times 10^{-3}$ | $5.0 \times 10^{-3}$ | $1.0 \times 10^{-4}$ | $1.0 \times 10^{-3}$ |
| BP | $1.0 \times 10^{-3}$ | $1.0 \times 10^{-4}$ | – | – | $1.0 \times 10^{-3}$ | $1.0 \times 10^{-4}$ |
| DBP | $1.0 \times 10^{-3}$ | $1.0 \times 10^{-3}$ | – | – | $1.0 \times 10^{-3}$ | – |

# E  Alignment as a function of noisy forward passes

Figure 8 shows the same data as Figure 1, but the $x$-axis displays the number of noisy forward passes performed, instead of the number of noise iterations. This figure essentially compares NP, ANP and INP given the same amount of computation.

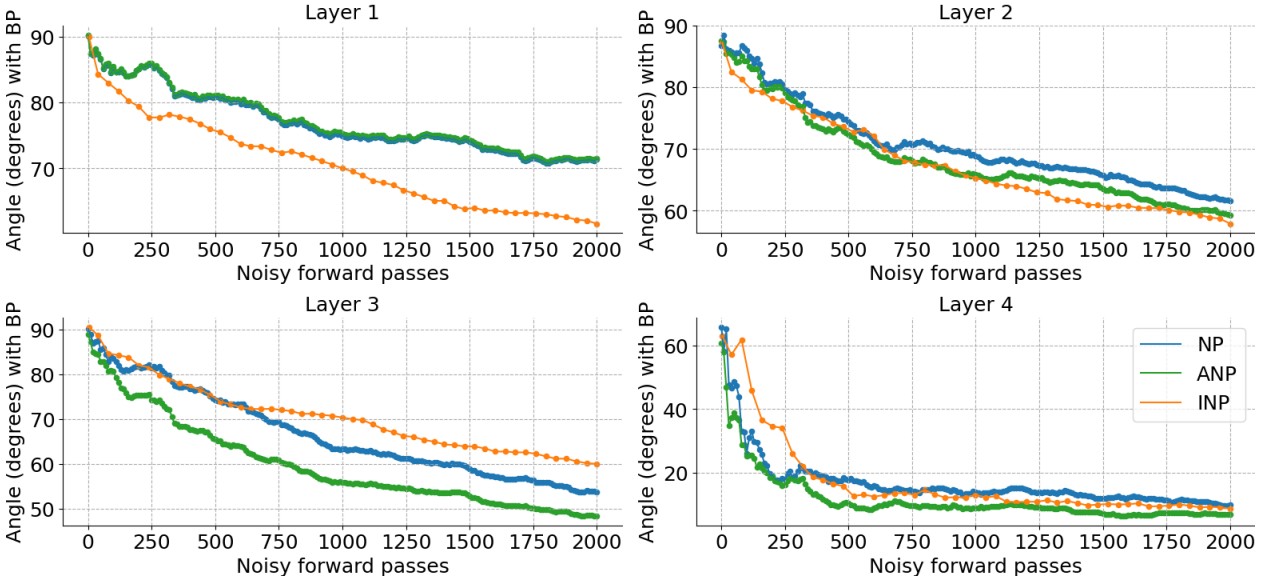

Figure 8: Angles between BP's weight update and an update calculated by NP, ANP and INP as a function of the number of noisy forward passes.

## F   Peak accuracies

Table 3 shows peak accuracies for all algorithms used in the CIFAR-10 experiments and Table 4 shows peak accuracies for the Tiny ImageNet experiment.

Table 3: Peak percentage accuracies for different learning algorithms and architectures on CIFAR-10. Best performance across all methods is shown in boldface.

| METHOD | 1 LAYER | | 3 HIDDEN LAYERS | | 6 HIDDEN LAYERS | | 9 HIDDEN LAYERS | | CONVNET | |
|---|---|---|---|---|---|---|---|---|---|---|
| | TRAIN | TEST | TRAIN | TEST | TRAIN | TEST | TRAIN | TEST | TRAIN | TEST |
| BP | 41.3 | 39.3 | 59.6 | 51.5 | – | – | – | – | **100.0** | **62.1** |
| DBP | **44.9** | **40.9** | **99.6** | **54.3** | – | – | – | – | 100.0 | 60.5 |
| NP | 38.5 | 38.2 | 28.1 | 28.1 | – | – | – | – | 45.2 | 44.5 |
| DNP | 42.8 | 40.2 | 44.9 | 42.1 | 46.7 | 43.1 | 40.5 | 38.6 | 50.2 | 47.9 |
| ANP | – | – | 30.4 | 30.6 | – | – | – | – | 52.7 | 49.9 |
| DANP | – | – | 48.5 | 44.4 | 47.3 | 44.2 | 44.8 | 42.7 | 58.2 | 52.6 |
| INP | – | – | 32.0 | 32.0 | – | – | – | – | 52.3 | 49.9 |
| DINP | – | – | 52.6 | 46.3 | **53.0** | **46.6** | **52.0** | **45.3** | 57.9 | 52.2 |

Table 4: Peak percentage accuracies for different learning algorithms on Tiny ImageNet. Best performance across all methods is shown in boldface.

| METHOD | TINY IMAGENET | |
|---|---|---|
| | TRAIN | TEST |
| BP | 55.5 | **14.5** |
| DANP 1 ITER | 2.2 | 2.1 |
| DANP 10 ITERS | 13.5 | 9.0 |
| DANP 30 ITERS | 32.8 | 11.3 |
| DANP 100 ITERS | 35.4 | 12.0 |
| DINP 1 ITER | 2.0 | 2.0 |
| DINP 10 ITERS | 11.4 | 8.1 |
| DINP 30 ITERS | 26.7 | 11.8 |
| DINP 100 ITERS | **60.8** | 13.4 |
| DNP 100 ITERS | 9.7 | 7.5 |

# G    Computational overhead of decorrelation

This section describes the theoretical amount of extra computation and memory that is required when applying decorrelation two a fully connected layer, a convolutional kernel and the amount of measured extra memory use in a three-layer fully connected network. Tables 5 and 6 display FLOPs and memory requirements for a fully connected layer and a convolutional layer, respectively. Table 7 shows measured memory consumption for a three-layer fully connected network for the various algorithms.

Table 5: Approximate FLOPS required for a forward pass through a fully connected $1000 \times 1000$ layer, batch size 1000, for BP, DBP, and DANP

| Algorithm | Approx. FLOPS | Brief Description of Calculation | Approx. Memory Required |
|---|---|---|---|
| **BP Total** | 6,000,000,000 | forward pass + backward pass + weight update $= 6 \times \text{batch\_size} \times \text{in\_dim} \times \text{out\_dim}$ | $\mathcal{O}\big(\text{batch\_size} \times (\text{in\_dim} + \text{out\_dim}) + \text{in\_dim} \times \text{out\_dim}\big)$ |
| **DBP Total** | 14,000,000,000 | forward $+ R \times z +$ backward $+ R^\top \times e +$ weight update $+ XX^\top +$ R update $= 6 \times \text{batch\_size} \times \text{in\_dim} \times \text{out\_dim} + 4 \times \text{batch\_size} \times \text{in\_dim}^2 + 2 \times \text{in\_dim}^3$ | $\mathcal{O}\big(\text{batch\_size} \times (\text{in\_dim} + \text{out\_dim}) + \text{in\_dim} \times \text{out\_dim} + \text{in\_dim}^2\big)$ |
| **DANP Total** | 14,000,000,000 | $2 \times (\text{forward} + R \times z) +$ weight update $+ XX^\top +$ R update $= 4 \times \text{batch\_size} \times \text{in\_dim} \times \text{out\_dim} + 4 \times \text{batch\_size} \times \text{in\_dim}^2 + 2 \times \text{in\_dim}^3$ | $\mathcal{O}\big(\text{batch\_size} \times (\text{in\_dim} + \text{out\_dim}) + \text{in\_dim} \times \text{out\_dim} + \text{in\_dim}^2\big)$ |

Table 6: Approximate FLOPS required for a forward pass through a convolutional kernel of $5 \times 5 \times 3 \times 8$ (H, W, chan_in, chan_out), batch size 1000, for BP, DBP, and DANP

| Algorithm | FLOPS | Brief description of calculation | Approx. Memory Required |
|---|---|---|---|
| BP Total | 3,600,000 | forward pass + backward pass (BP) + weight update $= 6 \times \text{batch\_size} \times \text{in\_dim} \times \text{out\_dim}$ | $\mathcal{O}\big(\text{batch\_size} \times (\text{in\_dim} + \text{out\_dim}) + \text{in\_dim} \times \text{out\_dim}\big)$ |
| DBP Total | 38,193,750 | forward $+ R \times z +$ backward (BP) $+$ decorrelation backward (same as forward) $+$ weight update $+ XX^\top +$ R update $= 6 \times \text{batch\_size} \times \text{in\_dim} \times \text{out\_dim} + 4 \times \text{batch\_size} \times \text{in\_dim}^2 + 2 \times \text{in\_dim}^3$ | $\mathcal{O}\big(\text{batch\_size} \times (\text{in\_dim} + \text{out\_dim}) + \text{in\_dim} \times \text{out\_dim} + \text{in\_dim}^2\big)$ |
| DANP Total | 38,193,750 | $2 \times (\text{forward} + R \times z) +$ weight update $+ XX^\top +$ R update $= 4 \times \text{batch\_size} \times \text{in\_dim} \times \text{out\_dim} + 4 \times \text{batch\_size} \times \text{in\_dim}^2 + 2 \times \text{in\_dim}^3$ | $\mathcal{O}\big(\text{batch\_size} \times (\text{in\_dim} + \text{out\_dim}) + \text{in\_dim} \times \text{out\_dim} + \text{in\_dim}^2\big)$ |

Table 7: Empirical memory use for parameters in MB in a 3-layer fully connected network trained on CIFAR-10

| Algorithm | Memory Use (MB) |
|---|---|
| BP | 20.04 |
| DBP | 68.04 |
| ANP | 20.04 |
| DANP | 68.04 |

