# OpenReview forum: "Node Perturbation Can Effectively Train Multi-Layer Neural Networks"
_TMLR — Rejected by TMLR_

### Review · Reviewer_PfUo · 2026-01-06

**Summary Of Contributions:**

Standard node perturbation (NP) is not very much data efficient and can be unstable. The paper develops a modern perspective on NP by relating it to the directional derivative and incorporating input decorrelation. It is found that a closer alignment with directional derivatives together with input decorrelation at every layer theoretically and practically leads to superior performance of NP learning with large improvements in parameter convergence and on the test data, approaching that of backpropagation.

**Audience:**

Yes

**Audience Explanation:**

I think so. The topic is definitely relevant to the audience of TMLR.

My main concern about the paper is that the contributions and novelties are not made that clear. For example, the author(s) argue that they introduce decorrelated node perturbation. You may think this is a novel method introduced in the paper when stated this way. However, it mentions the derivation of the procedure can be found in Ahmad et al. 2023. That makes it a bit confusing the novelty of the paper compared to Ahmad et al. 2023. It should be highlighted whether the paper introduces a novel decorrelated node perturbation method, or it is from the literature, but the paper provides some analysis using this decorrelation method.

Another main concern is that even though the numerical results show that decorrelation improves convergence and node perturbation can help  train the multi-layer networks, there seems a lack of discussions on why this is so. It will be helpful to include some discussions on the intuitions why they help with the performance.

**Claims And Evidence:**

Yes

**Claims Explanation:**

I think so. This is mostly an empirical paper. But there are also some simple derivations for the formulas that are provided in the appendix.

**Requested Changes:**

(1) At the end of page 2, please define $\tilde{a}_{k}$.

(2) On page 3, you have $\nabla_{v}\mathcal{L}=\lim_{h\rightarrow 0}\frac{\mathcal{L}(\tilde{x}_{L}(h))-\mathcal{L}(x_{L})}{h\Vert v\Vert}$. Note that before you take the limit $h\rightarrow 0$, the quantity $\frac{\mathcal{L}(\tilde{x}_{L}(h))-\mathcal{L}(x_{L})}{h\Vert v\Vert}$ is random. Therefore, you need to specify the meaning of the limit, e.g. whether it is almost sure convergence or convergence in distribution etc.

(3) In equation (2), please define what $N_{l}$ is. It seems later you use $N_{l}$ to denote the number of nodes in layer $l$.

(4) In equation (3), please define $\delta\mathcal{L}$. It seems you define it later on page 4. I think you should define it in equation (3) because it is the first time this notation appears.

(5) Right after equation (7), you wrote that $C_{kl}^{\mathrm{sgd}}$ is the covariance of SGD updates. Is there any equation for the SGD updates in the paper that you are referring to? If not, at least, you should explain what this SGD is about.

(6) In your equation (7), you have $\approx$. If it has been discussed in the literature when $\approx$ in equation (7) is a good approximation,  it will be helpful to add a sentence or two about that, and this might also help validate the numerical performance later on in the paper.

(7) In Section 2.1.1., you introduce iterative node perturbation; in Section 2.1.2, you introduce node perturbation; in Section 2.1.3, you introduce activity-based node perturbation. If the methodology exists in the literature, please provide some references. If the methodology is novel, and first appears in this paper, you should also highlight that.

---

### Review · Reviewer_Lzht · 2026-01-19

**Summary Of Contributions:**

The paper examines various node-perturbation (NP) techniques as alternatives to backpropagation (BP) for training neural networks. The main contribution lies in demonstrating the effectiveness of input decorrelation in improving the convergence and performance of NP-based approaches. However, the computational requirements of input decorrelation can be prohibitive and result in similar FLOPs and memory usage for ANP and BP.

**Additional Comments:**

N/A

**Audience:**

Yes

**Audience Explanation:**

Considering the related work discussed in this paper and the overall interest in NP techniques, the audience for this line of work might not be large in number. However, the paper presents input decorrelation as an effective and practical technique to improve NP, which currently suffers from stability and scalability challenges. This is an aspect that a few individuals may be interested in.

**Claims And Evidence:**

No

**Claims Explanation:**

- The formulations for directional derivatives, and the ideas about activity perturbations were already introduced in previous work, such as [1]. Thus, the claims must be adjusted to first discuss the related work and then clearly state the novel contributions.

- There are no baselines such as the forward AD [1] in all the experiments except the backpropagation approach. Is there a specific reason for this?

- Based on the compute requirements presented in Appendix G, the decorrelated BP and ANP require a similar amount of FLOPs and Memory. This requirement does not fit well with the overall message of the paper since the motivation was to propose alternatives to BP that can be suitable for resource-constrained hardware settings. These aspects lack an in-depth discussion.

- More importantly, the current results do not support the argument of using NP or even DNP techniques as an alternative to BP. If biological plausibility is an argument for NP, then a discussion is missing on this aspect.


[1] Ren, Mengye, et al. "Scaling Forward Gradient With Local Losses." The Eleventh International Conference on Learning Representations. 2023

**Requested Changes:**

1. What is the main difference between the directional derivative and activity perturbation formulations of this paper and the forward-mode AD and activity perturbed forward gradient introduced in [1]?

2. In Figure 1, how is the gradient for the BP baseline computed? What is the experimental setting, and which step is this gradient computed for? How does alignment change for different BP steps?

3. More importantly, this alignment does not seem to be necessarily useful since ANP methods outperform INP methods. Can the authors provide more insight into this?

4. A reference is missing in the caption of Figure 6.

---

### Review · Reviewer_sAAw · 2026-03-10

**Summary Of Contributions:**

This paper proposes a new _Node Perturbation_ (NP) training algorithm to train multi-layer neural networks.

As a reminder, NP consists in estimating the gradient of the loss w.r.t. the parameters by following the steps:
1. make a forward pass and compute the loss;
2. make a second forward pass with noise addition to one or several pre-activations, and compute the loss;
3. by considering the difference between the two losses and the noise added to each pre-activation, the gradient of the loss w.r.t. several parameters can be computed.

The contribution is two-fold:
1. the paper proposes the _Activity-based Node Perturbation_ (ANP), which approximates the gradient of the loss w.r.t. the parameters, not by using the _noise added_ in the pre-activations, but the _difference_ between the pre-activations obtained during the 2nd forward and the 1st forward passes. That way, the noise added in the first layers is taken into consideration when computing the gradient w.r.t. the parameters of the last layers.
2. the paper proposes a _decorrelation_ algorithm that removes progressively the correlations between the activations at every layer.

**Audience:**

Yes

**Audience Explanation:**

Proposing better training without backpropagation is an active field in deep learning that worth studying.

**Claims And Evidence:**

No

**Claims Explanation:**

There is a major omission in this paper: the proposed training algorithm (ANP) is not proven to be equivalent to a gradient descent (in any sense, e.g., in average over the noise), while preceding works on NP [1] take care of such theoretical properties.

This omission is major, not only for theoretical reasons, but also because considering the difference of pre-activations at a given node as noise is a bold claim, that may be false and lead to wrong results. In particular, this "noise" is not ensured to be centered, which is a vital (implicit) assumption in [1] to prove that NP is equivalent to gradient descent. In fact, we can even expect this noise _not to be centered_ at the end of training: at convergence, any (centered) noise injected in the first layer would move the output of the network towards the "wrong direction", which cannot be interpreted as centered noise whatsoever.

[1] _On the Stability and Scalability of Node Perturbation Learning_, Hiratani et al., 2022.

**Requested Changes:**

Add a formal proof that the proposed method ANP is equivalent to gradient descent descent in some sense. If not, add a discussion about this (negative) property and provide a heuristic to explain why it should be better than standard NP.

---

### Decision · Action_Editor_UDZZ · 2026-05-23

**Recommendation:** Reject

**Audience:**

Yes

**Audience Explanation:**

The paper studies an important and active topic: alternatives to backpropagation for training neural networks, with relevance to biologically plausible learning, forward-only training, and neuromorphic or noisy hardware systems. However, the current version requires stronger theoretical characterization and clearer positioning before its findings can be considered sufficiently mature for publication.

**Claims And Evidence:**

No

**Claims Explanation:**

The submission addresses an interesting and relevant problem, and the empirical results suggest that decorrelation can substantially improve node perturbation style training. However, the reviews identified unresolved issues in the theoretical justification and positioning of the method. In particular, the proposed ANP update is not sufficiently characterized as a gradient descent like estimator, the novelty relative to forward-gradient/activity-perturbation methods and prior decorrelation work is not clearly delineated, and the computational overhead of decorrelation weakens the resource-constrained hardware motivation. The paper contains promising ideas, but the current evidence and framing do not yet support the main claims.

**Resubmission Of Major Revision:**

The authors may consider submitting a major revision at a later time.

---

> ### Author Response · Authors · 2026-05-26
> **Revised version**
>
> Dear Action editor,
>
> We were just about to upload a revised version of the paper, including new appendices and some additional baseline comparisons, along with rebuttals. Could you please consider the revised version of the paper before making a definitive decision? We are ready to upload later today.
>
> The authors